# Socioeconomic and ethnical disparity in coronary heart disease outcomes in Denmark and the effect of cardiac rehabilitation—A nationwide registry study

**Ingunn Kjesbu** [1]*, **Eva Prescott**[1], **Hanne Rasmusen H. K.**[1], **Merete Osler**[2,3], **Mogens Lytken Larsen**[4], **Ida Gustafsson**[1], **Ann Dorthe Zwisler**[5,6], **Kirstine Laerum Sibilitz**[7]

1 Department of Cardiology, Bispebjerg Frederiksberg University Hospital, Copenhagen, Denmark, 2 Centre for Clinical Research and Disease Prevention, Bispebjerg Frederiksberg University Hospital, Copenhagen, Denmark, 3 Section of Epidemiology, Department of Public Health, University of Copenhagen, Copenhagen, Denmark, 4 Department of Clinical Medicine, Aalborg University, Aalborg, Denmark, 5 REHPA, The Danish Knowledge Centre for Rehabilitation and Palliative Care, Odense University Hospital, Nyborg, Denmark, 6 Department of Clinical Research, University of Southern Denmark, Odense, Denmark, 7 Department of Cardiology, Copenhagen University Hospital, Rigshospitalet, Copenhagen, Denmark

* ingunn.eklo.kjesbu.01@regionh.dk

**Data Availability Statement:** All relevant data are within the manuscript and its Supporting information files.

## Abstract

### Aims

Cardiovascular patients with low socioeconomic status and non-western ethnic background have worse prognostic outcomes. The aim of this nationwide study was first to address whether short-term effects of hospital-based outpatient cardiac rehabilitation (CR) are similar across educational level and ethnic background, and secondly to study whether known disparity in long-term prognosis in patients with cardiovascular disese is diminished by CR participation.

### Methods

All patients with myocardial infarction and/or coronary revascularization from August 2015 until March 2018 in the Danish national patient registry or the Danish cardiac rehabilitation database (DHRD) were included. We used descriptive statistics to address disparity in achievement of quality indicators in CR, and Cox proportional hazard regression to examine the association between the disparity measures and MACE (cardiovascular hospitalization and all-cause mortality) with adjustment for age, gender, index-diagnose and co-morbidity.

### Results

We identified 34,511 patients of whom 19,383 had participated in CR and 9,882 provided information on CR outcomes from the DHRD. We demonstrated a socioeconomic gradient in improvements in $VO_{2peak}$, and non-western patients were less often screened for depression or receive dietary consulting. We found a strong socioeconomic gradient in MACE irrespective of CR participation, medication, and risk factor control (adjusted HR 0.65 (95% CI

**Funding:** The authors received no specific funding for this work.

**Competing interests:** The authors have declared that no competing interests exist.

0.56–0.77) for high versus low education). Non-western origin was associated with higher risk of MACE (adjusted HR 1.2 (1.1–1.4)).

## Conclusion

We found only minor socioeconomic and ethnic differences in achievement of CR quality indicators but strong differences in CHD prognosis indication that conventional risk factor control and medical treatment following CR do not diminish the socioeconomic and ethnical disparity in CHD prognosis.

## Introduction

Cardiac rehabilitation (CR) aims to limit the physiological and psychological distress of coronary heart disease (CHD), to optimize medical treatment, and to decrease mortality and morbidity for patients with CHD [1]. Lack of proper follow up after acute myocardial infarction (MI) or in patients with stable CHD who are revascularized might diminish the effect of the initial treatment. CR is shown to improve morbidity and mortality in patients with CHD and has received a class IA recommendation by the European Society of Cardiology and the American College of Cardiology [2, 3]. However, cardiovascular patients with low socioeconomic status and non-western origin less often participate in CR, and also have worse prognostic outcomes [4–9]. Thus, CR underutilization and lower adherence to secondary prevention among cardiovascular patients with low socioeconomic status and non-western origin could affect the disparities, and we hypothesize that CR have the potential to diminish the disparities in prognostic outcomes following CHD.

In this nationwide study we aimed first to determine whether short-term effects of CR are similar across socioeconomic status and ethnic backgrounds, and secondly to study whether disparity in the long-term prognosis of patients with cardiovascular disease is diminished by CR participation.

## Methods

### Setting

The Danish health-care system is tax funded, and access to health-care services is free of charge for those holding a Danish citizenship or a long-term residency permit. Further, cost of prescribed medicine is partly covered. CR in Denmark consist of three phases. Phase I is the immediate phase in-hospital after event. Phase II refers to the initial 8–12 weeks of outpatient CR at hospitals and municipalities. Phase III is the consecutive follow-up and maintenance phase in primary care [10]. In this article, we focus on hospital-based phase II CR.

### Study population

We included all patients discharged alive from hospitals in Denmark hospitalized with first or recurrent MI and/or stable CHD revascularized from August 2015 until March 2018 identified in the Danish National Patient Register (DNPR) [11] with the following diagnoses (following International Classification of Diseases, 10th revision codes (ICD-10 codes)): DI20-DI25, including acute coronary syndrome (ST-elevation MI, Non ST-elevation MI) as primary or secondary diagnosis (I210-I219, I248, I240, I249) and stable angina pectoris as primary diagnosis (I209, I251, I251B, I251C). Furthermore, the Nordic Classification of Surgical Procedures

(SKS-codes) were used to identify revascularization as percutaneous coronary intervention (PCI), and coronary artery bypass grafting (CABG) according to the DHRD definition (KFNA, KFNB, KFNC, KFNG, KFNW, KFNF, KFNH, KFNJ, KFNK, KFNW).

A subgroup of CR participants was identified in the Danish Cardiac Rehabilitation Database (DHRD). The two databases were merged and assessed through Statistics Denmark [12]. The overlap with DNPR is depicted in supplemental Figure 1 in S1 File. DHRD is a Danish online, national quality improvement database on CR aimed at patients with CHD. The database was implemented in 2015 and approved by the Danish Health and Medicine Authorities [13]. Reporting to the database is mandatory for all hospitals in Denmark delivering phase II CR, and the database includes 11 quality indicators reflecting intermediate outcomes in CR. Data are registered for patients hospitalized for MI or stable CHD treated with PCI, CABG, or medication alone, and data are collected at patient level as well as CR program level. Information on demographic data, main cardiovascular risk factors at baseline and at end-of-CR, medication, and diagnostic tests are recorded directly to the database or gathered through linkage with The Danish National Heart Register. In 2018–2019, the database covered 57% of all CR participants nationwide ranging between 25–75% in the different CR centers [14].

## Socioeconomic status and ethnicity

The main predictors explored in this study were socioeconomic status measured as level of education and ethnicity. Information about educational level was retrieved from the National Education Register. Patients were divided into groups based on the highest attained educational level registered at the time of inclusion: basic education (primary school), short education (secondary school, trade/craft education) medium education (high school, bachelor, ≤3 years), and high education (university (MBA, post-doc, master, >3 years)).

Patients were stratified by ethnic background into "Danish"/ "Western"/ "non-Western" according to data from The Danish Civil registration system [15]. The patient was classified as being Danish if at least one parent held a Danish citizenship and was born in Denmark. Western patients were defined as patients from the EU, Andorra, Australia, Canada, Iceland, Liechtenstein, Monaco, New Zealand, Norway, San Marino, Switzerland, Great Britain, United States of America, or the Vatican City. All other patients were defined as non-Western patients.

## Outcomes

For the first aim the outcomes used to measure short-term effects of CR were the following quality indicators in the DHRD [13]: 1. ≥80% participation in training sessions; 2. ≥10% increase in $VO_{2peak}$; 3. smoking cessation if smoker at baseline; 4. received dietary consulting; 5. achieved LDL goal (LDL<1.8 mmol/L or 50% decrease in LDL level); 6. achieved blood pressure goal (<140/90 mmHg); 7. screened for diabetes if no prior history of diabetes; 8: screened for depression (hospital anxiety and depression score–subscale for depression (HADS-D)) [16]; 9.-11. medical treatment with antiplatelets or anticoagulants, statins, and betablockers at end-of-CR.

In the second aim the long-term prognostic outcome was defined as the composite endpoint of hospitalization for CHD and all-cause mortality (MACE). CHD hospitalization was defined as readmission due to acute MI (I21), unstable angina pectoris (I20), heart failure (I50), or stroke (I63) through linkage to the DNPR. Information on vital status used for all-cause mortality was retrieved from The Danish Civil registration system [15]. MACE was defined as occurring at least 28 days after index event.

**Covariables.** Covariables retrieved from the DNPR included age, gender, type of index-event (MI, stable angina), revascularization (CABG, PCI), CR participation (if the course of disease was connected to any CR procedure code including physical activity, psychological intervention, dietary consulting, and out-patient follow up) [14], previous CHD, co-morbidity according to weighted Charlson comorbidity index (wCI) [17], co-habitation status (living with spouse/child or living alone), work status (employee, unemployed/early retirement, retiree) and income. Known hypertension, hypercholesterolemia and diabetes were defined as treatment with relevant drugs prior to index event and the information was retrieved from the prescription register [18]. Data on smoking-status and comorbidity (cancer, peripheral artery disease, stroke, chronic obstructive pulmonary disease, kidney disease, depression, muscular disease) were retrieved from the DHRD for the registered CR participants.

## Statistical analyses

First, baseline characteristics and quality indicators were analyzed using descriptive statistics and compared across strata defined by education and ethnicity. We used Pearson's chi-squared test for categorical variables, one-way *ANOVA* for normally distributed continuous variables and Kruskall-Wallis test for non-normally distributed continuous variables.

Secondly, we used Cox proportional hazard regression (hazard ratio (HR) and 95% confidence intervals (CI)) to determine the association between educational level and ethnicity and the risk of MACE. The patients were followed from index event until recurrent event, death, emigration, or at end of follow up if they were event free. Follow-up ended on December 31st, 2018. The results were presented by Kaplan Meier survival plots. CR participation was included as a time-dependent covariable in the Cox proportional hazard model. The proportional hazard assumption was examined by using Schönfeld residuals [19] implemented in the estat phtest command in STATA. All models were adjusted for gender, age, type of index-event and co-morbidity (previous CHD, hypertension, hypercholesterolemia, diabetes and wCI). Effect modification was assessed as a second step in the Cox model by applying a model assuming effect modification (interaction) between educational level or ethnicity and CR. This model was tested against the crude analyses by a likelihood ratio test. To account for the potential lag-time to effect of CR, the Cox proportional hazard analyses with CR as time-dependent covariable (or potential effect modifier) was repeated by using a Landmark approach where the follow-up was started at 6-months after inclusion. All patients were included in the analyses unless they had died during the first 6 months after event. All analyses were performed using STATA 16.1 (StataCorp, Texas, USA). A p-value below 0.05 was considered statistically significant.

## Ethics

The study was approved by the Danish Data Protection agency (j.nr.: 2012-58-0004). According to Danish law no written consent on patient level was needed in this study.

## Results

### Socioeconomic and ethnic disparity in CR quality indicators

**The Danish Cardiac Rehabilitation Database (DHRD) population.** The DHRD population comprised 9,882 patients participating in CR. The majority, 77%, had basic or short education while 6% had high education. Patients with basic education were older, had lower proportion of males compared to short, medium, and high education, more often lived alone, had more depression and comorbidity, and were less often treated with PCI and CABG

**Table 1. Demographic characteristics, socioeconomic factors, and comorbidity by education among 9,882 patients in the DHRD population.**

| Variable | | Basic | Short | Medium | High | p-value |
|---|---|---|---|---|---|---|
| N total 9,882 | | 3,149 (32%) | 4,474 (45%) | 1,655 (17%) | 585 (6%) | |
| Age, mean (SD) | | 66.2 (11.5) | 62.2 (10.8) | 64.6 (10.7) | 64.4 (10.7) | <0.001 |
| Gender | Male | 2118 (67.3) | 3520 (78.7) | 1203 (72.7) | 501 (85.6) | <0.001 |
| MI | | 1451 (46.1) | 2099 (46.9) | 746 (45.1) | 269 (44.4) | 0.470 |
| PCI | | 1120 (35.6) | 1716 (38.4) | 622 (37.6) | 239 (40.9) | 0.026 |
| CABG | | 314 (10.0) | 496 (11.1) | 183 (11.1) | 92 (15.7) | <0.001 |
| Socioeconomic factors | | | | | | |
| Living alone | | 1129 (35.9) | 1235 (27.6) | 436 (26.4) | 158 (27.2) | <0.001 |
| Work status | Self-employed | 115 (3.7) | 174 (3.9) | 50 (3.0) | 42 (7.2) | <0.001 |
| | Employee | 684 (21.7) | 1569 (35.1) | 628 (37.9) | 258 (44.2) | |
| | Unemployed/early retirement | 506 (16.1) | 421 (9.4) | 121 (7.3) | 29 (5.0) | |
| | Retiree | 1796 (57.0) | 2267 (50.7) | 833 (50.3) | 249 (42.6) | |
| Income (quartiles) | 0–25 | 944 (30.0) | 953 (21.3) | 175 (10.6) | 52 (8.9) | <0.001 |
| | 25–50 | 865 (27.5) | 942 (21.1) | 229 (13.8) | 48 (8.2) | |
| | 50–75 | 773 (24.5) | 1120 (25.0) | 458 (27.7) | 83 (14.2) | |
| | 75–100 | 567 (18.0) | 1459 (32.6) | 793 (47.9) | 401 (68.7) | |
| CHD risk factors | | | | | | |
| Hypertension | | 2360 (74.9) | 3270 (73.1) | 1156 (69.8) | 378 (64.6) | <0.001 |
| Hypercholesterolemia | | 2295 (72.9) | 3194 (71.4) | 1113 (67.3) | 353 (60.3) | <0.001 |
| Diabetes | | 578 (18.4) | 713 (15.9) | 217 (13.1) | 74 (12.6) | <0.001 |
| Known CHD | | 903 (28.7) | 1168 (26.1) | 430 (26.0) | 118 (20.2) | <0.001 |
| Smoking status | Never | 959 (30.7) | 1464 (32.9) | 668 (40.7) | 289 (50.0) | <0.001 |
| | Former | 1509 (48.2) | 2229 (50.0) | 777 (47.3) | 232 (40.1) | |
| | Active | 660 (21.1) | 761 (17.1) | 196 (11.9) | 57 (9.9) | |
| Comorbidities | | | | | | |
| Cancer | | 67 (2.1) | 131 (2.9) | 48 (2.9) | 28 (4.8) | 0.064 |
| PAD | | 95 (3.0) | 108 (2.4) | 40 (2.4) | 8 (1.4) | 0.005 |
| Stroke | | 180 (5.7) | 213 (4.8) | 82 (5.0) | 27 (4.6) | 0.146 |
| COPD | | 237 (7.5) | 200 (4.5) | 69 (4.2) | 16 (2.7) | <0.001 |
| Kidney disease | | 76 (2.4) | 90 (2.0) | 33 (2.0) | 20 (3.4) | 0.191 |
| Depression | | 318 (10.1) | 372 (8.3) | 121 (7.3) | 37 (6.3) | 0.005 |
| Muscle /skeletal | | 408 (13.0) | 469 (10.5) | 135 (8.2) | 29 (5.0%) | <0.001 |
| Charlson wCI, mean (SD) | | 1.47 (1.38) | 1.38 (1.41) | 1.33 (1.42) | 1.27 (1.27) | <0.001 |

Data are presented as N (%) unless otherwise specified. Abbreviations: SD: Standard deviation, CR: Cardiac rehabilitation, MI: acute myocardial infarction, PCI: percutaneous coronary intervention, CABG: coronary artery bypass graft, CHD: coronary heart disease, PAD: peripheral artery disease, COPD: chronic obstructive pulmonary disease, wCI: weighted Charlson comorbidity index

(Table 1). The majority of the population was of Danish origin (90%). Compared to Western and non-Western, the Danish patients were older and more often female. Non-western patients suffered from more cardiovascular risk factors, more depression, and were more often on early retirement (Table 2).

**Achievement of quality indicators in CR.** We found a statistically significant gradient with education in achievement of $VO_{2peak}$ goal ($\geq$10% increase in $VO_{2peak}$) ranging from 43% among those with basic education to 53% in those with high education (p = 0.006). Smoking cessation also differed significantly across education groups (p<0.001), but with the lowest quit rate among those with highest education. Non-Western patients less often received

**Table 2. Demographic characteristics, socioeconomic factors and comorbidity by ethnicity among 9,882 patients in the DHRD population.**

| Variable | | Denmark | Western | Non-western | p-value |
|---|---|---|---|---|---|
| N total 9882 | | 8928 (90%) | 328 (4%) | 614 (6%) | |
| Age, mean (SD) | | 65.5 (10.9) | 63.6 (10.9) | 57.6 (10.6) | <0.001 |
| Gender | Male | 6600 (73.9) | 247 (75.3) | 499 (81.3) | <0.001 |
| MI | | 4167 (46.7) | 151 (46.0) | 245 (39.9) | 0.005 |
| PCI | | 3338 (37.4) | 130 (39.6) | 235 (38.3) | 0.660 |
| CABG | | 961 (10.8) | 32 (9.8) | 89 (14.5) | 0.013 |
| Socioeconomic factors | | | | | |
| Living alone | | 2626 (29.4) | 140 (42.7) | 198 (32.2) | <0.001 |
| Work status | Self-employed | 343 (3.8) | 14 (4.3) | 24 (3.9) | <0.001 |
| | Employee | 2872 (32.2) | 106 (32.3) | 156 (25.4) | |
| | Unemployed/early retirement | 797 (8.9) | 41 (12.5) | 244 (39.7) | |
| | Retiree | 4835 (54.2) | 155 (47.3) | 262 (26.4) | |
| Income (quartiles) | 0–25 | 1822 (20.4) | 90 (27.4) | 218 (35.5) | <0.001 |
| | 25–50 | 1837 (20.6) | 66 (20.1) | 186 (30.3) | |
| | 50–75 | 2229 (25.0) | 91 (27.7) | 114 (18.6) | |
| | 75–100 | 3040 (34.1) | 81 (24.7) | 96 (15.6) | |
| CHD risk factors | | | | | |
| Hypertension | | 6540 (73.3) | 219 (66.8) | 412 (67.1) | 0.180 |
| Hypercholesterolemia | | 6314 (70.7) | 216 (65.9) | 433 (70.5) | <0.001 |
| Diabetes | | 1351 (15.1) | 36 (11.0) | 198 (32.2) | <0.001 |
| Known CHD | | 2340 (26.2) | 82 (25.0) | 204 (33.2) | <0.001 |
| Smoking status | Never | 3046 (34.3) | 108 (33.5) | 226 (37.4) | 0.354 |
| | Former | 4381 (49.3) | 141 (43.8) | 228 (37.7) | |
| | Active | 1454 (16.4) | 73 (22.7) | 151 (25.0) | |
| Comorbidities | | | | | |
| Cancer | | 250 (2.8) | 11 (3.4) | 13 (2.1) | 0.005 |
| PAD | | 224 (2.5) | 12 (3.7) | 16 (2.6) | 0.752 |
| Stroke | | 451 (5.1) | 26 (7.9) | 25 (4.1) | 0.679 |
| COPD | | 473 (5.3) | 21 (6.4) | 29 (4.7) | 0.956 |
| Kidney disease | | 190 (2.1) | 10 (3.0) | 20 (3.3) | 0.961 |
| Depression | | 752 (8.4) | 25 (7.6) | 70 (11.5) | 0.003 |
| Muscle/skeletal | | 967 (10.8) | 27 (8.2) | 52 (8.5) | 0.359 |
| Charlson wCI, mean (SD) | | 1.41 (1.41) | 1.37 (1.36) | 1.26 (1.25) | 0.040 |

Data are presented as N (%) unless otherwise specified. Abbreviations: SD: Standard deviation, CR: Cardiac rehabilitation, MI: acute myocardial infarction, PCI: percutaneous coronary intervention, CABG: coronary artery bypass graft, CHD: coronary heart disease, PAD: peripheral artery disease, COPD: chronic obstructive pulmonary disease, wCI: weighted Charlson comorbidity index

dietary consulting (p = 0.002) and were less often screened for depression (p<0.001). No difference in initiation of post-MI medical treatment or conventional risk factor control following CR was found across level of education and ethnicity (Fig 1a and 1b and Supplementary Table 2a, 2b in S1 File).

## Socioeconomic and ethnic disparity in prognostic outcome

**The Danish National Patient Registry (DNPR) population.** A total of 36,325 patients was identified in the DNPR with MI or revascularization. 1,814 patients had died in hospital or within the first 28 days after index event, thus 34,511 patients were included in the study. Of

### a.  by education

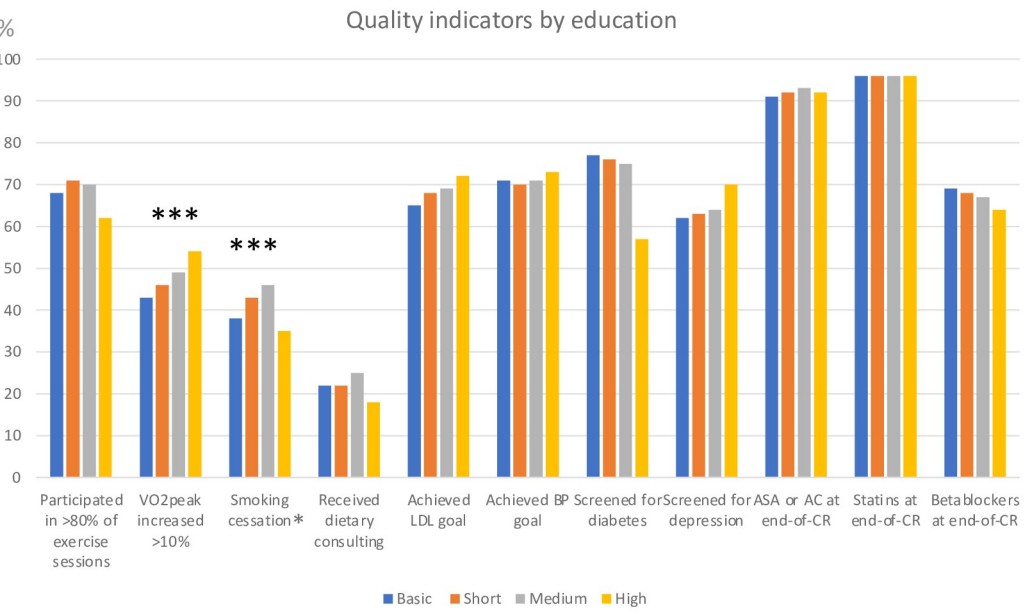

### b.  by ethnicity

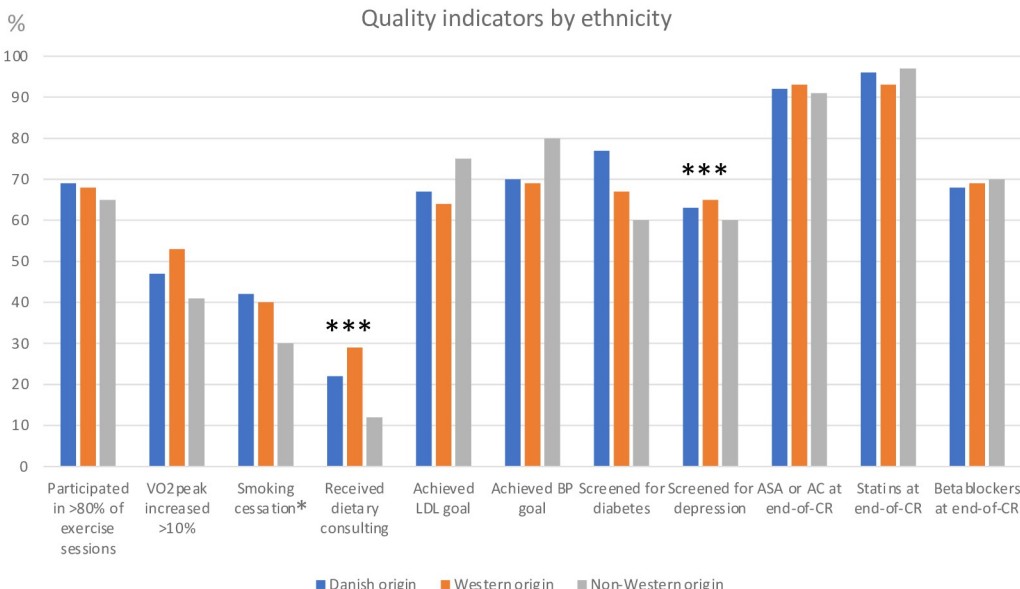

**Fig 1. Achievement of quality indicators in CR across education and ethnicity among 9.882 patients in the DHRD population.** a. by education and b. by ethnicity. *** p-value <0.05, adjusted for gender, age, comorbidity index, diabetes, hypertension, hypercholesterolemia and previous CHD; *smokers at baseline. Abbreviations: $VO_{2peak}$: peak oxygen consumption, LDL: Low density lipoprotein, BP: blood pressure, ASA; acetylsalicylic acid, AC: anticoagulants, CR: cardiac rehabilitation.

**Table 3. Demographic characteristics, socioeconomic factors, and comorbidity CR participation among 34.511 patients in the DNPR population.**

| Variable | | CR | No CR | p-value |
|---|---|---|---|---|
| N total 34,511 | | 19,383 (56%) | 15,128 (44%) | |
| Age, mean (SD) | | 67.4 (12.0) | 67.4 (13.1) | 0.460 |
| Gender | Male | 12993 (72.6) | 9533 (68.5) | <0.001 |
| MI | | 10123 (53.0) | 9125 (47.0) | <0.001 |
| PCI | | 8824 (56.0) | 6954 (44.0) | 0.410 |
| CABG | | 3583 (97.0) | 115 (3.0) | <0.001 |
| Socioeconomic factors | | | | |
| Education | Basic | 6417 (56.5) | 4947 (43.5) | 0.009 |
| | Short | 7569 (57.6) | 5573 (42.4) | |
| | Medium | 2544 (57.6) | 1869 (42.4) | |
| | High | 806 (53.5) | 701 (46.5) | |
| Ethnicity | Danish | 16278 (57.7) | 11932 (42.3) | <0.001 |
| | Western | 469 (49.0) | 488 (51.0) | |
| | Non-western | 913 (47.0) | 1029 (53.0) | |
| Living alone | | 6162 (54.5) | 5137 (45.5) | <0.001 |
| Work status | Self-employed | 596 (57.0) | 446 (43.0) | <0.001 |
| | Employee | 4563 (57.0) | 3452 (43.0) | |
| | Unemployed/early retirement | 1988 (53.3) | 1739 (46.7) | |
| | Retiree | 10334 (57.5) | 7653 (42.5) | |
| Income (quartiles) | 0–25 | 4434 (56.8) | 3373 (43.2) | 0.019 |
| | 25–50 | 4428 (56.7) | 3378 (43.3) | |
| | 50–75 | 4359 (55.8) | 3447 (44.2) | |
| | 75–100 | 4494 (57.6) | 3312 (42.4) | |
| CHD risk factors | | | | |
| Hypertension | | 13314 (57.8) | 9697 (42.2) | <0.001 |
| Hypercholesterolemia | | 11338 (59.4) | 7764 (40.6) | <0.001 |
| Diabetes | | 3455 (58.0) | 2493 (42.0) | 0.002 |
| Known CHD | | 7929 (56.0) | 6178 (44.0) | 0.900 |
| Comorbidity | | | | |
| Charlson wCI, mean (SD) | | 1.77 (1.65) | 1.93 (1.63) | <0.001 |

Data are presented as N (%) unless otherwise specified. Abbreviations: SD: Standard deviation, CR: Cardiac rehabilitation, MI: acute myocardial infarction, PCI: percutaneous coronary intervention, CABG: coronary artery bypass graft, CHD: coronary heart disease, wCI: weighted Charlson comorbidity index

these, 19,383 (56%) participated in CR (Supplementary Figure 1 in S1 File). CR participants were more often male whereas age did not differ between the CR participants and non-participants (Table 3).

Participation rate was 53.5%, among patients with high education compared to approximately 57% in the remaining groups defined by level of education. Across ethnicity, CR participation rate was higher among Danish patients (57%) compared to Western- and non-Western patients (49% and 47%, respectively). CR participants had more diabetes and hypertension but had lower comorbidity score. MI as index diagnose was less frequent and CABG more frequent among CR participants.

**Prognostic outcome by CR participation.** A total of 8,575 patients (25%) registered in DNPR experienced MACE during a median (min/max) follow-up of 1.9 years (0.02/3.4 years), of whom 3,676 (43%) died. There was a clear gradient of lower risk of MACE with higher level of education reaching a HR of 0.61 (.53-.67) in those with the highest education with basic

**Table 4. Risk of MACE across level of education and ethnicity among 19,383 CR participants assessed by hazard ratios (95% confidence intervals) from Cox-regression models.**

| By education | Crude HR (95% CI) | p-value | Adjusted HR (95% CI)* | p-value |
|---|---|---|---|---|
| Basic | 1 (base) | | | |
| Short | 0.75 (0.72–0.79) | <0.001 | 0.85 (0.80–0.90) | <0.001 |
| Medium | 0.69 (0.64–0.74) | <0.001 | 0.84 (0.77–0.92) | 0.001 |
| High | 0.60 (0.53–0.67) | <0.001 | 0.65 (0.56–0.77) | <0.001 |
| By ethnicity | | | | |
| Danish | 1 (base) | | | |
| Western | 0.98 (0.87–1.1) | 0.806 | 1.1 (0.94–1.2) | 0.217 |
| Non-western | 0.95 (0.86–1.1) | 0.236 | 1.2 (1.1–1.3) | 0.001 |

*adjusted for age, gender, comorbidity index, diabetes, hypertension, hypercholesterolemia, previous CHD, index-event and CR as a time-dependent variable.

Abbreviations: HR: Hazard ratio; CI: Confidence interval

education as reference (Table 4, Fig 2a). The association was attenuated by adjustment (for age, gender, index-event, wCI, diabetes, hypertension, hypercholesterolemia, and previous CHD) but remained statistically significant. The gradient was similar in those participating and those not participating in CR, with no evidence of effect modification (p = 0.367) (Supplementary Table 4 in S1 File). Danish and Western patients had similar risk of MACE whereas adjusted HR (95% CI) for MACE was 1.2 (1.1–1.4) in non-Western (Table 4, Fig 2b).

The analyses were repeated using a Landmark approach where follow-up was started at 6 months after inclusion. A total of 939 patients had died before the 6-months follow-up landmark. The gradient in the risk of MACE across level of education remained persistent, whereas the association between ethnicity and MACE was attenuated and no longer remained statistically significant.

## Discussion

In this nationwide cohort study, using data from Danish registers to investigate socioeconomic and ethnic disparity and short-term effects of CR in CHD patients in Denmark, we found that among participants in CR, increasing level education was associated with greater improvement in $VO_{2peak}$ and non-western patients were less likely to be screened for depression or receive dietary consulting. We found strong socioeconomic and ethnic disparity in prognosis with seemingly little impact of CR participation, medical treatment, or conventional risk factor control at end-of-CR to diminish this disparity.

### Socioeconomic disparity in coronary heart disease

A recently published report from the Danish National Board of Health showed that the socioeconomic disparity in CHD events and mortality has been increasing for the last 10 years [20]. This "welfare paradox" of persisting or increasing health inequalities in highly developed western countries has been described as one of the great disappointments in modern public health [21]. This inequity also applies to CR and secondary prevention. A recently published European study (the EU-CaRE study) on 1,700 elderly CHD patients from 7 different countries found that education was associated with poorer risk factor control and lower exercise capacity before CR, and that this disparity either persisted or increased following CR [6]. This higher risk factor burden post-MI could lead to higher mortality and morbidity at long-term, however our study indicates a strong association between educational level and prognostic outcome following CHD despite similar medication and risk factor control following CR. This is in line

a. by education

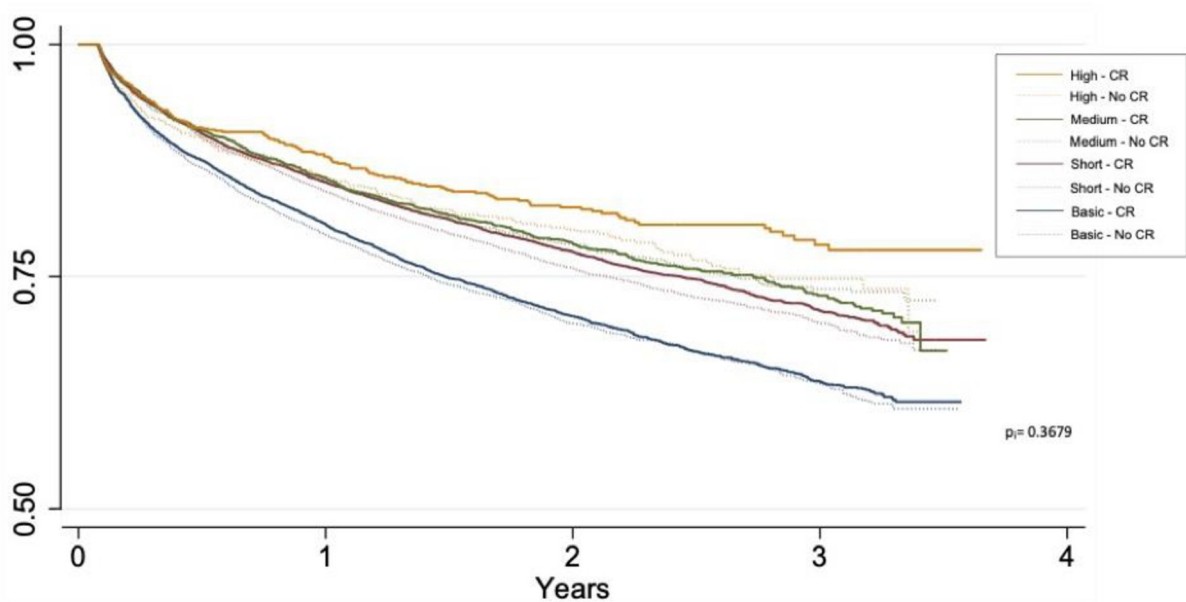

b. By ethnicity

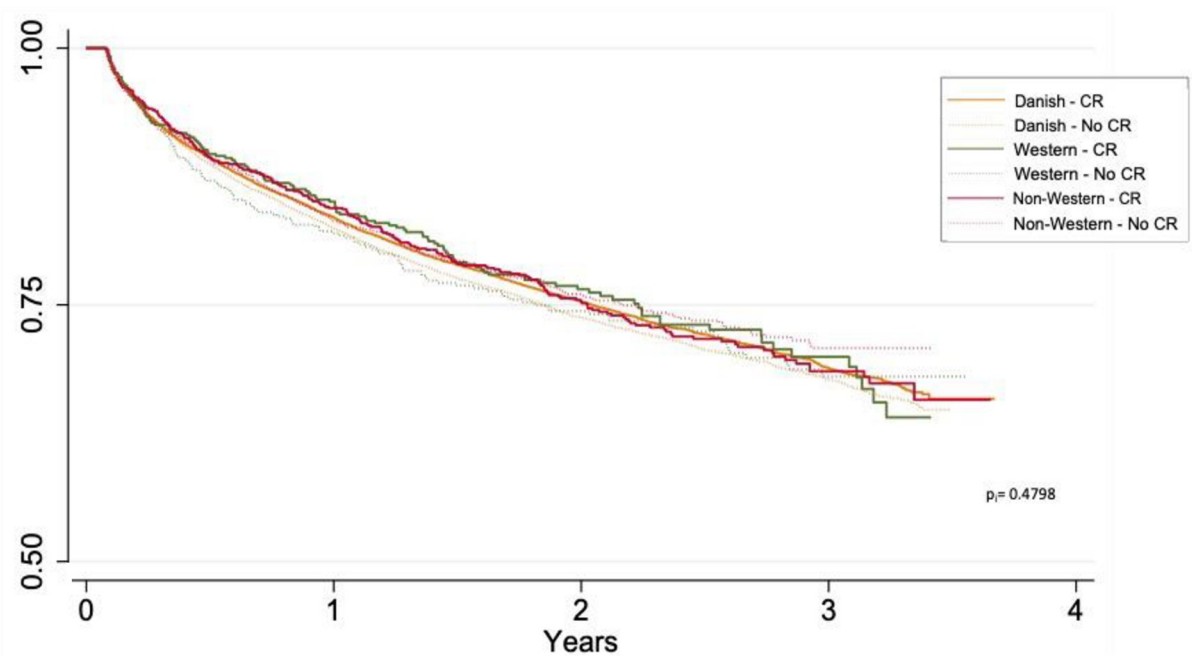

**Fig 2. Risk of MACE by CR participation across education and ethnicity among 34.511 patients in the DNPR population presented by Kaplan-Meier plots.** a. by education and b. By ethnicity. $p_i$ = p for interaction, abbreviations: CR: Cardiac rehabilitation.

with a small study that found persisting socioeconomic disparity in long-term outcome (all-cause mortality, cardiovascular mortality and non-fatal cardiovascular events) despite improved CHD risk factor control and improved adherence to medical treatment in socioeconomic deprived patients at short-term [22]. Health behavior is established early in life and further influenced by conditions during childhood. The lower socioeconomic status, the more risk factors a person will be exposed to and accumulate throughout life also called "clustering of health determinants" [23]. From early life, factors influencing health are for example social factors that affect early cognitive, emotional, and social development. Later in life, education, income, occupation, life-style related risk factors such as smoking, alcohol, diet, and physical activity are of great importance. One might then assume that at the point of life when a patient suffers a MI, important factors and exposures accumulated during life beyond conventional cardiovascular risk factors will also highly influence the morbidity and mortality outcome. This could explain why although we see similar achievement of risk factor control and post-MI medication, this is not sufficient to attenuate socioeconomic disparity in CHD outcome.

## Ethnical disparity in coronary heart disease

Studies in Northern European countries concerning migrants´ use of secondary prevention after CHD are few and mostly on small populations. A Swedish nationwide study from 2015 demonstrated no significant differences between migrants and Swedish-born CHD patients in the use of recommended drugs after MI [24]. This is in line with our study, indicating that no ethnic disparity in initiation of recommended medication after MI. However, a nationwide study on Danish CHD patients from 2017 showed that non-Western migrants had higher discontinuation of all recommended medications after MI (failure to claim reimbursement of a new prescription within 90 days after previous prescription). In agreement with our study, they also found that non-western patients had fewer contacts for physical exercise and less often received dietary advise post-MI [7]. In addition, we found that non-western patients less often were screened for depression, however the explanation for this could be that not all centres offer screening tools in other languages than Danish and English.

Nonetheless, lower adherence to secondary prevention at long term among migrants could have great consequences in mortality and morbidity. In this study, we found that non-Western patients are at greater risk of MACE following MI/revascularization. Although this is in line with several studies demonstrating increased mortality among migrants compared to local born [25, 26], other studies have found that migrants are not always worse off [27–29]. These conflicting results are probably due the categorization of a heterogenous populations in terms of country of origin, cultural and genetic differences. A pan-European study comparing mortality rates among migrants after CHD in 6 different European countries found mortality rates to be higher among migrants from some regions (Sub-Saharan Africa, South Asia, North Africa and Turkey) and lower in migrants from other regions (East Asia and Latin America) [30]. With this in mind, the results from this study cannot be generalized to all non-Western CHD patients, however, focus on adherence to secondary prevention at long-term seems pivotal to address and overcome ethnical disparity in CHD outcome.

## Clinical implications and further research

Based on the results in this study and knowledge from previous studies, several suggestions could be included to diminish socioeconomic and ethnical disparity in CHD patients in future clinical practice. Focus after MI and in general in CHD patients with low socioeconomic status should be beyond conventional CHD risk factors with focus on general health behaviour, and more intense coaching with health education to improve adherence to CR and understanding

the patients' needs. Further, psychological and lifestyle interventions, non-pharmacological treatment as well as medical treatment at long-term should be delivered bearing in mind that traditional risk factors might not necessarily be the barrier for failing with CR in patients with other ethnic background than Western or socioeconomic deprived patients. Further research should probably focus on which cultural and innate health behaviours that affect CHD outcome and how these can be addressed, as well as identifying barriers to treatment adherence especially in patients with non-Western background.

## Strengths and limitations

By using nationwide registers, we were able to include all eligible patients and to include data on number of contacts to health services, diagnosis, and deaths with complete follow-up. As the data is collected prospectively, there is no risk of information bias.

However, the variables in the databases used in this study were not chosen for our research question. Consequently, it was not possible to address all potential confounding variables making the study prone to unmeasured confounding. Furthermore, we were not able to do individual analysis on center level. However, all analysis were adjusted for center, However, we cannot rule out that the findings in this study is affected by differences in CR programs at center level (i.e. areas with high proportion of patients with low SES have less resources in CR programs). As previously described, the Danish health care system is free of charge and parts of the medical costs is covered. Thus, the results from this study might not be generalizable to countries where access to health care services differs in the population (e.g. across socioeconomic groups).

Another limitation is the coverage of the DHRD database. The database was recently established and the coverage in some centres is still low (range 25–75%). Furthermore, the data completeness of some variables is poor (Supplementary Table 2 in S1 File). Some patients are referred directly to CR in municipal care (e.g. non-Danish speaking patients in some areas), and unfortunately data on phase II CR conducted in the municipalities are not yet available to the database. This may lead to selection bias and the results can be difficult to interpret and to generalize to all CR centres and all CHD patients in Denmark.

Finally, a limitation of this study is the classification of CR participants in the DNPR database and the overlap of the DHRD database. This overlap is depicted in Supplementary Figure 2 in S1 File. Although great effort was taken to identify all CR participants by using a wide range of CR codes, there is still a chance of misclassification and drift in classification in register diagnoses and procedures over time which could lead to type 2 error.

## Conclusion

In this nationwide, register based cohort study, we found few differences in CR participation, initiation of medical treatment and conventional risk factor control after CR across socioeconomic and ethnic groups. We confirmed a strong socioeconomic and ethnic differences in CHD prognosis with limited impact of CR. Further research should focus on which cultural and innate health behaviours that affect CHD outcome, and how these can be addressed to achieve socioeconomic and ethnical equality in patients with cardiovascular disease.

## Supporting information

**S1 File.**
(DOCX)

## Acknowledgments

We thank our colleagues for the data collection in DHRD.

## Author Contributions

**Conceptualization:** Ingunn Kjesbu, Eva Prescott, Merete Osler, Mogens Lytken Larsen, Ida Gustafsson, Ann Dorthe Zwisler, Kirstine Laerum Sibilitz.

**Formal analysis:** Ingunn Kjesbu.

**Methodology:** Eva Prescott, Merete Osler, Ann Dorthe Zwisler, Kirstine Laerum Sibilitz.

**Supervision:** Eva Prescott, Kirstine Laerum Sibilitz.

**Validation:** Eva Prescott.

**Visualization:** Ingunn Kjesbu.

**Writing – original draft:** Ingunn Kjesbu.

**Writing – review & editing:** Ingunn Kjesbu, Eva Prescott, Hanne Rasmusen H. K., Merete Osler, Mogens Lytken Larsen, Ida Gustafsson, Ann Dorthe Zwisler, Kirstine Laerum Sibilitz.

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
