## [Decision Letter · Decision Letter 0]

20 Jun 2022

PONE-D-22-13483Socioeconomic and ethnical disparity in coronary heart disease outcomes in Denmark and the effect of cardiac rehabilitation – a nationwide registry studyPLOS ONE

Dear Dr. Kjesbu,

Thank you for submitting your manuscript to PLOS ONE. After careful consideration, we feel that it has merit but does not fully meet PLOS ONE’s publication criteria as it currently stands. Therefore, we invite you to submit a revised version of the manuscript that addresses the points raised during the review process. Thank you for submitting your paper to PLOS One. Your study is interesting; please consider reviewer comments and submit a revised version of your manuscript.

We look forward to receiving your revised manuscript.

Kind regards,

Salil Deo

Academic Editor

PLOS ONE

Journal Requirements:

Additional Editor Comments:

Please revise based on reviewer comments

Reviewers' comments:

Reviewer's Responses to Questions

**Comments to the Author**

1. Is the manuscript technically sound, and do the data support the conclusions?

Reviewer #1: Yes

Reviewer #2: Yes

2. Has the statistical analysis been performed appropriately and rigorously? 

Reviewer #1: I Don't Know

Reviewer #2: Yes

3. Have the authors made all data underlying the findings in their manuscript fully available?

Reviewer #1: Yes

Reviewer #2: Yes

4. Is the manuscript presented in an intelligible fashion and written in standard English?

Reviewer #1: Yes

Reviewer #2: Yes

5. Review Comments to the Author

Reviewer #1: The authors are to be commended for this analysis of the impact of Phase 2 Cardiac Rehabilitation on the care of patients with CVD in Denmark. "The data is the data"--and the authors appropriately note limitations inherent in their retrospective analysis utilizing 2 large, but imperfect and overlapping data bases.

Two items warrant attention:

Internal validity-The primary unit of analysis is the individual patient. It is probable that the individual CR programs differ substantially in their patient population (e.g. lower vs. higher SES), resources (some programs may be better resourced), and accessibility. It would be appropriate to analyze results from a program perspective, to see if there were disparities in process or outcome variable based upon program attended, and how this compared with individual patient characteristics. This may also provide actionable information regarding CR program improvement.

External validity-This study provides an analysis of patients in the single country of Denmark. A brief description of how these results may or may not be generalizable to other countries (i.e. particular characteristics of the Danish population and health system) would strength the discussion and conclusion.

Reviewer #2: Thanks for the opportunity to review the original article submitted by

Kjesbu et al. In this analysis of a large cohort of patients with myocardial infarction or coronary revascularization the Danish national registries, the investigators show that cardiac rehabilitation participation was overall low, and its components varied by patient demographic and economic position. The authors also found association between patient socioedemographics and major adverse cardiovascular event risk.

Abstract: well written overall.

Introduction

• Page 5/Line 8 – “American Association of Cardiology” should be “American College of Cardiology”

• Avoid the use of “cardiac patients”. Should be replaced with “patients with cardiovascular disease”

• Line 16 – “backgrounds”

Methods

- ICD10 codes usually has one letter and followed by number. The authors seem to have “D” in front of the ICD10 code. Are these Danish specific codes?

- Page 6/Line 16 – typo “DI219”

- Reference is needed for the Nordic procedure codes.

- The authors mention that reporting is mandatory for all CR patients to the DHRD, but in page 7/line 10-11, they report “In 2018-2019, the database covered 57% of all CR participants nationwide ranging between 25-75% in the different CR centers”. Can the authors explain the discrepancy?

- Can the authors explain what countries were defined as “Western”?

- Perhaps a better term than ethnicity used here is “nativity”.

Results

- Page 12/paragraph 1 should be the first paragraph in the results section.

Discussion

- Overall well written. No comments

Figures/Tables

- Consider adding variance around the bar graph estimates (95% Confidence intervals).

- Consider adding adjusted estimates with logistic regressions and 95% confidence interval for CR quality indicators.

- Consider adding % of patients achieving all CR quality indicators by nativity and education.

6. PLOS authors have the option to publish the peer review history of their article (what does this mean?). If published, this will include your full peer review and any attached files.

Reviewer #1: **Yes: **Richard A Josephson, MS MD

Reviewer #2: **Yes: **Sadeer Al-Kindi

---

## [Author Response · Author response to Decision Letter 0]

9 Aug 2022

Response to Reviewers 

Authors' comment: We thank both editors for excellent feedback and editing suggestions. 

Please see the answers to the questions below.

5. Review Comments to the Author

Reviewer #1: The authors are to be commended for this analysis of the impact of Phase 2 Cardiac Rehabilitation on the care of patients with CVD in Denmark. "The data is the data"--and the authors appropriately note limitations inherent in their retrospective analysis utilizing 2 large, but imperfect and overlapping data bases.

Two items warrant attention:

Internal validity-The primary unit of analysis is the individual patient. It is probable that the individual CR programs differ substantially in their patient population (e.g. lower vs. higher SES), resources (some programs may be better resourced), and accessibility. It would be appropriate to analyze results from a program perspective, to see if there were disparities in process or outcome variable based upon program attended, and how this compared with individual patient characteristics. This may also provide actionable information regarding CR program improvement.

Author response: Thank you for this comment. We do agree that there might be substantial differences in the patient populations on how the CR programs are utilized and how it is offered for several reasons. Ideally, we wanted to examine this more thoroughly in the DHRD database, however the data was not good enough to conclude on this matter. We do believe that this is an important issue of further investigation, and that more research is needed.

External validity-This study provides an analysis of patients in the single country of Denmark. A brief description of how these results may or may not be generalizable to other countries (i.e. particular characteristics of the Danish population and health system) would strength the discussion and conclusion.

Author response: We do agree that the results might not be generalizable to all populations / countries. Thus, the setting of this study is described in the methods section, and we have elaborated further on this issue on page 17, line 18-21. 

Reviewer #2: Thanks for the opportunity to review the original article submitted by

Kjesbu et al. In this analysis of a large cohort of patients with myocardial infarction or coronary revascularization the Danish national registries, the investigators show that cardiac rehabilitation participation was overall low, and its components varied by patient demographic and economic position. The authors also found association between patient sociodemographic and major adverse cardiovascular event risk.

Abstract: well written overall.

Introduction

• Page 5/Line 8 – “American Association of Cardiology” should be “American College of Cardiology”

Authors response: This has now been edited

• Avoid the use of “cardiac patients”. Should be replaced with “patients with cardiovascular disease”

Author response: This has now been adjusted throughout the manuscript when found appropriate.

• Line 16 – “backgrounds”

Authors response: This has now been edited

Methods

- ICD10 codes usually has one letter and followed by number. The authors seem to have “D” in front of the ICD10 code. Are these Danish specific codes?

Author response: The D has appropriately been removed. The codes are the international ICD10 codes.

- Page 6/Line 16 – typo “DI219”

Author response: This is now corrected 

- Reference is needed for the Nordic procedure codes.

Author response: Citation added.

- The authors mention that reporting is mandatory for all CR patients to the DHRD, but in page 7/line 10-11, they report “In 2018-2019, the database covered 57% of all CR participants nationwide ranging between 25-75% in the different CR centers”. Can the authors explain the discrepancy?

Author response: Although the reporting is mandatory, it took some years for the centres to integrate and implement this system. This could be due to several factors both on center (systematic) and a personal level. The coverage has been described in the “strengths and limitations” section (page 17, line 11-17)

- Can the authors explain what countries were defined as “Western”?

Author response: This includes the following countries: EU, Andorra, Australia, Canada, Iceland, Liechtenstein, Monaco, New Zealand, Norway, San Marino, Switzerland, Great Britain, USA, and Vatican City.

This has been added in the manuscript page 7, line 22-25.

- Perhaps a better term than ethnicity used here is “nativity”.

Authors response: We have considered different terms to be used in the manuscript, as one should be careful when dividing the patients into ethnical groups. We have after thorough consideration landed on the term “ethnicity” with a more thorough description of what we mean by using this term (country of origin) in the methods section. 

Results

- Page 12/paragraph 1 should be the first paragraph in the results section.

Author response: In the results section we decided to describe the two databases separately as they do not perfectly overlap. This is to limit the confusion about the overlap, and to clarify which database describes the analysis of quality indicators and the analysis of the prognostic outcome. This setup of the results section has been thoroughly discussed in the author's group. Paragraph 1 of page 12 describes the secondary/prognostic outcome (The DNPR database).

Discussion

- Overall well written. No comments

Figures/Tables

- Consider adding variance around the bar graph estimates (95% Confidence intervals).

- Consider adding adjusted estimates with logistic regressions and 95% confidence interval for CR quality indicators.

Author response: Table of quality indicators with numbers and the adjusted p-values are included in the supplemental data.

- Consider adding % of patients achieving all CR quality indicators by nativity and education.

Author response: Due to varying data completeness of the variables (described in supplementary data) we find that a graph like this could be misleading and have chosen not to include this analysis.

---

## [Decision Letter · Decision Letter 1]

23 Aug 2022

PONE-D-22-13483R1Socioeconomic and ethnical disparity in coronary heart disease outcomes in Denmark and the effect of cardiac rehabilitation – a nationwide registry studyPLOS ONE

Dear Dr. Kjesbu,

Thank you for submitting your manuscript to PLOS ONE. After careful consideration, we feel that it has merit but does not fully meet PLOS ONE’s publication criteria as it currently stands. Therefore, we invite you to submit a revised version of the manuscript that addresses the points raised during the review process.

We look forward to receiving your revised manuscript.

Kind regards,

Salil Deo

Academic Editor

PLOS ONE

Journal Requirements:

Additional Editor Comments:

Thank you for your changes. We would like to accept your manuscript after you consider replying to the point raised by reviewer #1. Once you submit a revised version with that point taken care of, we can accept your manuscript. Thank you.

Reviewers' comments:

Reviewer's Responses to Questions

**Comments to the Author**

1. If the authors have adequately addressed your comments raised in a previous round of review and you feel that this manuscript is now acceptable for publication, you may indicate that here to bypass the “Comments to the Author” section, enter your conflict of interest statement in the “Confidential to Editor” section, and submit your "Accept" recommendation.

Reviewer #1: (No Response)

Reviewer #2: All comments have been addressed

2. Is the manuscript technically sound, and do the data support the conclusions?

Reviewer #1: Partly

Reviewer #2: Yes

3. Has the statistical analysis been performed appropriately and rigorously? 

Reviewer #1: I Don't Know

Reviewer #2: Yes

4. Have the authors made all data underlying the findings in their manuscript fully available?

Reviewer #1: Yes

Reviewer #2: Yes

5. Is the manuscript presented in an intelligible fashion and written in standard English?

Reviewer #1: Yes

Reviewer #2: Yes

6. Review Comments to the Author

Reviewer #1: Please see prior comments. It is uncertain if the findings are due to variation in individual patients vs. variation in CR programs (i.e. programs serving lower SES patients may have less programmatic staffing or other resources). If this analysis is not part of the manuscript, then this limitation should be explicitly described.

Reviewer #2: Thank you for submitting the revised version.

The authorsn have addressed all my comments and the other reviewers comments

7. PLOS authors have the option to publish the peer review history of their article (what does this mean?). If published, this will include your full peer review and any attached files.

Reviewer #1: **Yes: **Richard Josephson

Reviewer #2: **Yes: **Sadeer Al-Kindi

---

## [Author Response · Author response to Decision Letter 1]

3 Oct 2022

Thank you for your last feedback and comment. 

6. Review Comments to the Author

Reviewer #1: Please see prior comments. It is uncertain if the findings are due to variation in individual patients vs. variation in CR programs (i.e. programs serving lower SES patients may have less programmatic staffing or other resources). If this analysis is not part of the manuscript, then this limitation should be explicitly described.

Authors response: A section about this limitation has been added, page 17, line 10-14

Both this response and the previous response to review is attached to the document.

---

## [Editor Report · Decision Letter 2]

13 Oct 2022

Socioeconomic and ethnical disparity in coronary heart disease outcomes in Denmark and the effect of cardiac rehabilitation – a nationwide registry study

PONE-D-22-13483R2

Dear Dr. Kjesbu,

We’re pleased to inform you that your manuscript has been judged scientifically suitable for publication and will be formally accepted for publication once it meets all outstanding technical requirements.

Kind regards,

Salil Deo

Academic Editor

PLOS ONE

Additional Editor Comments (optional):

Thank you for submitting your work to PLOS One and addressing all comments made by reviewers.
---

## [Editor Report · Acceptance letter]

26 Oct 2022

PONE-D-22-13483R2 

Socioeconomic and ethnical disparity in coronary heart disease outcomes in Denmark and the effect of cardiac rehabilitation – a nationwide registry study 

Dear Dr. Kjesbu:

I'm pleased to inform you that your manuscript has been deemed suitable for publication in PLOS ONE. Congratulations! Your manuscript is now with our production department. 

Kind regards, 

on behalf of

Dr. Salil Deo 

Academic Editor

PLOS ONE